

# Generalized Linear Models outperform commonly used canonical analysis in estimating spatial structure of presence/absence data

Lélis A. Carlos-Júnior[1,2,3], Joel C. Creed[4], Rob Marrs[2], Rob J. Lewis[5], Timothy P. Moulton[4], Rafael Feijó-Lima[1,6] and Matthew Spencer[2]

[1] Programa de Pós-Graduação em Ecologia e Evolução, Universidade do Estado do Rio do Janeiro, Rio de Janeiro, Brazil
[2] School of Environmental Sciences, University of Liverpool, Liverpool, United Kingdom
[3] Departamento de Biologia, Pontifícia Universidade Católica do Rio de Janeiro, Rio de Janeiro, Brazil
[4] Departamento de Ecologia, Universidade do Estado do Rio de Janeiro, Rio de Janeiro, Brazil
[5] Department of Forest Genetics and Biodiversity, Norwegian Institute of Bioeconomy Research, Bergen, Norway
[6] Division of Biological Sciences, University of Montana, Missoula, MT, United States of America

Corresponding author
Lélis A. Carlos-Júnior,
lelisjr_cjr@puc-rio.br,
lelisufmg@gmail.com

## ABSTRACT

**Background**. Ecological communities tend to be spatially structured due to environmental gradients and/or spatially contagious processes such as growth, dispersion and species interactions. Data transformation followed by usage of algorithms such as Redundancy Analysis (RDA) is a fairly common approach in studies searching for spatial structure in ecological communities, despite recent suggestions advocating the use of Generalized Linear Models (GLMs). Here, we compared the performance of GLMs and RDA in describing spatial structure in ecological community composition data. We simulated realistic presence/absence data typical of many $\beta$-diversity studies. For model selection we used standard methods commonly used in most studies involving RDA and GLMs.

**Methods**. We simulated communities with known spatial structure, based on three real spatial community presence/absence datasets (one terrestrial, one marine and one freshwater). We used spatial eigenvectors as explanatory variables. We varied the number of non-zero coefficients of the spatial variables, and the spatial scales with which these coefficients were associated and then compared the performance of GLMs and RDA frameworks to correctly retrieve the spatial patterns contained in the simulated communities. We used two different methods for model selection, Forward Selection (FW) for RDA and the Akaike Information Criterion (AIC) for GLMs. The performance of each method was assessed by scoring overall accuracy as the proportion of variables whose inclusion/exclusion status was correct, and by distinguishing which kind of error was observed for each method. We also assessed whether errors in variable selection could affect the interpretation of spatial structure.

**Results**. Overall GLM with AIC-based model selection (GLM/AIC) performed better than RDA/FW in selecting spatial explanatory variables, although under some simulations the methods performed similarly. In general, RDA/FW performed unpredictably, often retaining too many explanatory variables and selecting variables associated with incorrect spatial scales. The spatial scale of the pattern had a negligible effect on
GLM/AIC performance but consistently affected RDA's error rates under almost all scenarios.

**Conclusion**. We encourage the use of GLM/AIC for studies searching for spatial drivers of species presence/absence patterns, since this framework outperformed RDA/FW in situations most likely to be found in natural communities. It is likely that such recommendations might extend to other types of explanatory variables.

## INTRODUCTION

Ecological communities tend to be spatially structured in response to environmental gradients that are themselves organized in space, or to spatially contagious processes such as growth, dispersion, and species interactions (*Legendre & Legendre, 2012*; *Peres-Neto & Legendre, 2010*). Thus, disentangling the causes of spatial structure and identifying spatial variability and different scales of organization in natural communities is a central question in ecology (*Legendre, 1993*). Answering this question requires the construction of explanatory variables based on spatial relationships among sites (*Dray, Legendre & Peres-Neto, 2006*). One approach extensively used to create spatial variables and/or control for spatial autocorrelation in residuals is an eigenvector-based method, called Moran's eigenvector maps (MEMs, *Dray, Legendre & Peres-Neto, 2006*). This method creates spatial explanatory variables representing structure on a range of spatial scales from the spatial relationships among sampling sites. These variables can be used for a broad range of goals, from controlling for phylogenetic autocorrelation in ecological data (*Diniz-Filho et al., 2012*) to searching for spatial structure in natural communities, even when irregularly sampled (e.g., *Bauman et al., 2016*; *Neves et al., 2015*).

In many studies the response variables for which ecologists seek to find spatial structure are community composition datasets containing either abundances or presence/absence information (here, we focus on the latter). For community ecology studies, Redundancy Analysis (RDA) is one of the most popular strategies due to its versatile framework, well-established literature and abundant toolkits available for implementation (see (*Blanchet et al., 2014*) ; *Borcard, Legendre & Drapeau, 1992*; *Eisenlohr & De Oliveira-Filho, 2015*; *Saiter et al., 2015*). The RDA algorithm searches for optimal linear combinations (in the least-squares sense, see *Legendre & Legendre, 2012*) of the explanatory variables that best explain the variation in the transformed community composition data (*Legendre & Gallagher, 2001*; *Borcard, Gillet & Legendre, 2011*; (*Blanchet et al., 2014*)). The usual approach then consists of establishing the global significance of the relationship between the response matrix and all the explanatory variables, after which a subset of explanatory variables is usually selected by stepwise procedure such as Forward Selection (FW, sensu *Blanchet, Legendre & Borcard, 2008a*) The most common approach uses two thresholds for variable selection: a significance level $\alpha$ and the adjusted $R^2$ (see below and *Blanchet, Legendre &*

*Borcard, 2008a* for details). This whole framework will hereafter be called RDA/FW for brevity. A statistic related to the Akaike Information Criteria (AIC, *Akaike, 1973*) has also been suggested for RDA model selection (*Godínez-Domínguez & Freire, 2003*), but it has been shown to perform poorly and will not be further explored here (*Bauman et al., 2018a*).

However, methods based on least-squares such as RDA are unlikely to perform well when applied to data that violate the assumption of constancy in the mean–variance relationship. This assumption is usually violated by datasets containing many zeros including abundance (count or semi-quantitative) and presence/absence (binary) data. Data transformation does not always solve such problems (*O'Hara & Kotze, 2010*; *Warton, 2018*), although least-squares can give reasonably robust tests of the significance of regression coefficients (*Ives, 2015*). In general, algorithmic methods such as RDA do not take into account the statistical properties of the response variable, such as the distribution of variances and how the response changes along spatial/environmental gradients (*Ferrier et al., 2007*; *Warton, Wright & Wang, 2012*; *Warton et al., 2015*; *Warton, 2018*). More recently, Generalized Linear Models (GLMs) have been proposed as an alternative model-based approach to the analysis of presence/absence or count data (*Wang et al., 2012*; *Warton et al., 2015*; *Yee, 2006*). The use of GLMs has long been established for univariate analyses and related approaches for multivariate count data are now available (*O'Hara & Kotze, 2010*; *Warton, 2018*). The usual approach to selection of explanatory variables in this approach is Akaike's Information Criterion (AIC: *Akaike, 1973*; *Wagenmakers & Farrell, 2004*). This framework will hereafter be named GLM/AIC.

Here, we compared the performance of the RDA/FW and GLM/AIC approaches to selecting spatial explanatory variables for community presence/absence data by measuring the proportion of spatial patterns contained in simulated communities they could correctly retrieve. There have been some studies of simulated multivariate count data (*Warton, Wright & Wang, 2012*), but presence/absence data are particularly important in spatial studies because they are often the only data that can be collected consistently over large spatial extents. We therefore compare the performance of RDA/FW and GLM/AIC methods for the selection of MEM spatial variables (including one special case, the asymmetric eigenvector maps or AEM) from realistic simulated presence/absence data. We used spatial variables as our predictors since we were interested in discovering whether varying the spatial scales in which communities were structured would affect model performance. We generated simulated data sets with predefined spatial structure based on three real data sets, under two different ecological interpretations of presence/absence data. First, we assumed that species are truly present at some sites and absent at others, and are detected if present (simulated presence method, SPM). Alternatively, absences may represent failure to detect species that are truly present. In this case, we simulated species abundances, followed by a simulated sampling step to obtain presence/absence data (simulated abundances method, SAM).

## MATERIALS & METHODS

### Baseline datasets

We compared the two approaches to spatial variable selection using simulated community data based on three real community composition datasets with a range of properties:

1. Presence/Absence of 110 marine benthic macroalgae species from a Rapid Assessment Program for biodiversity of 42 sample sites spanning roughly 2,000 km$^2$ at Ilha Grande Bay, Rio de Janeiro, Brazil (southwest Atlantic) (*Carlos-Júnior et al., 2019*), permit number IBAMA/RJ:031/04);

2. Presence/Absence of 588 plant species from grassland covering 500 km$^2$ of Scotland's coast. Data were collected from 3639 5× 5 m quadrats from 94 sites. We used sites as our sample units, treating species as present when they occurred in at least one quadrat at a site, and absent otherwise (see *Lewis, Pakeman & Marrs, 2014*) for more information);

3. Presence/Absence of 47 freshwater aquatic insect species collected from 30 sample sites in five tributaries of the Guapiaçú River basin, Brazil which covers about 40 km$^2$ (Feijó-Lima in prep, permit number INEA-RJ: 019-2014).

For each of the datasets we used the geographical coordinates (maps and sampling sites in Fig. S1) to calculate spatial explanatory variables for regression (Fig. 1). We chose MEMs as our spatial variables since they are commonly used to describe spatial structure in ecological studies. Moreover, in contrast to coarser methods such as trend-surface analysis, MEMs are a flexible method, capable of describing all spatial scales provided by the sampling design (*Borcard, Gillet & Legendre, 2011*). They are also more flexible and powerful than the method of principal coordinates of neighbor matrices (PCNMs, a special case of distance-based MEMs) (*Bauman et al., 2018a*; *Bauman et al., 2018b*; *Borcard & Legendre, 2002*; *Dray, Legendre & Peres-Neto, 2006*). One needs two matrices to build the MEM variables for a given set of site coordinates: matrix **B** describing the connectivity among the geographical sampling sites and matrix **A** describing the weights of such connections. The Hadamard product of these two matrices generates the spatial weighting matrix (matrix **W**), which is then doubly centred and diagonalized, yielding eigenvectors to be used as spatial variables. For ecological studies, the processes of interest are usually those generating positive autocorrelation, and it is therefore common to use only MEMs associated with positive eigenvalues (as in this study). For studies in which negative spatial autocorrelation is also of interest (e.g., where negative interactions such as competitive exclusion, predation, etc are suspected), the eigenvectors associated with negative eigenvalues can also be separately used (*Bauman et al., 2018a*). We made decisions about **B** and **A** for each dataset based on our ecological knowledge of the spatial structure of these regions, since our goal was to simulate communities with ecologically sensible spatial structures. Therefore, for dataset 1 we chose the minimum spanning tree (**B**) with Euclidian linear distances as weights (**A**). Our decision was based on the shape of the bay and the fact that the main water movements make the sampling sites geographically compartmentalised in subregions where sites are likely to be minimally connected (*Carlos-Júnior et al., 2019*). Similarly, spatial organisation in dataset 2 could be sensibly described in terms of Delaunay

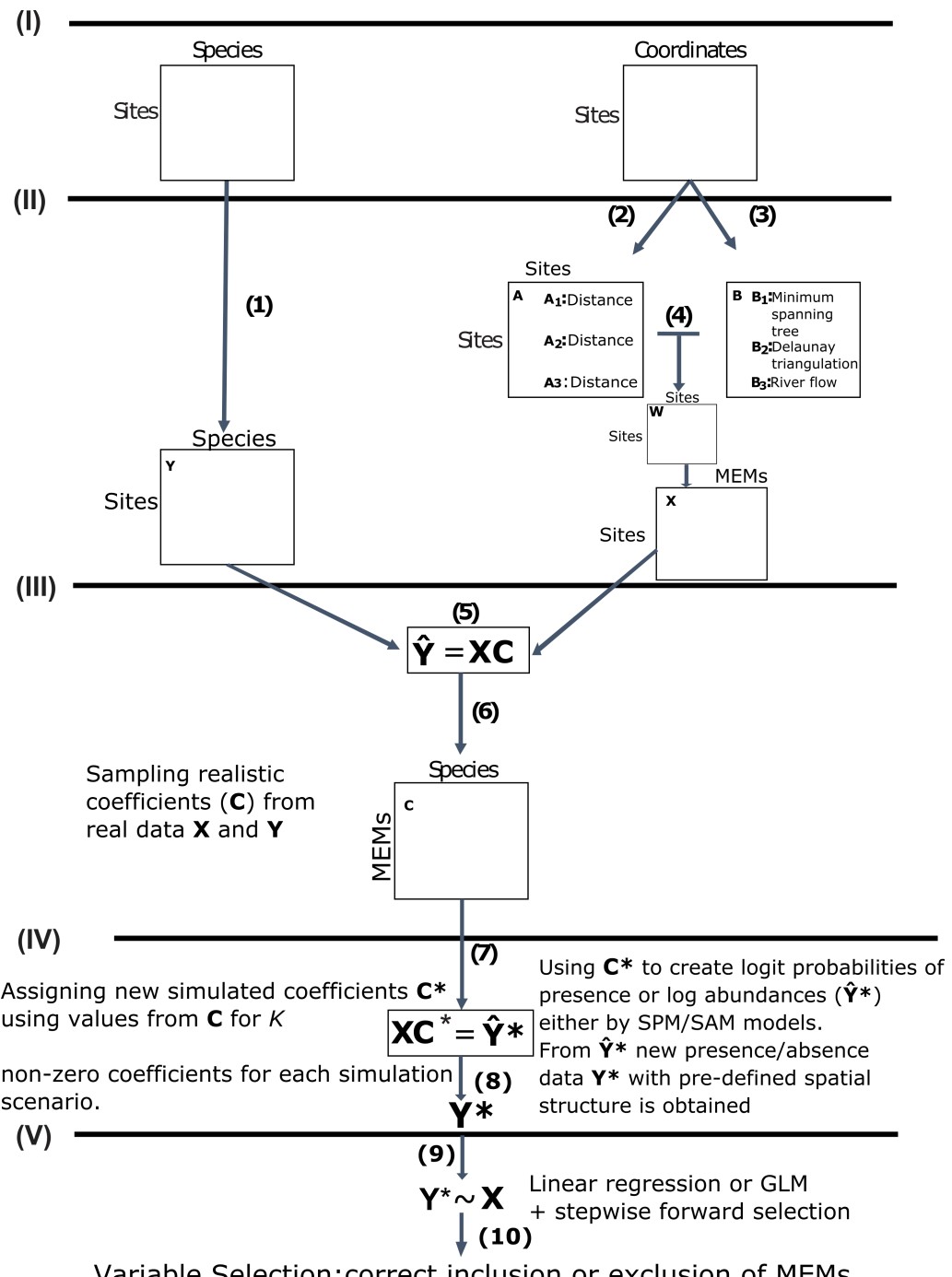

**Figure 1** **Schematic diagram of the main steps used in this study to simulate community presence/absence data with pre-defined spatial structure. Data acquisition (I):** We used real data from marine, terrestrial and freshwater communities and their respective sampling site coordinates as our baseline datasets. **Obtaining response and predictor matrices (II):** Those datasets were used to construct a response matrix of presence/absence data **Y** (1) and a matrix **X** of spatial explanatory variables called MEMs. The spatial variables were obtained from a pairwise site-by-site distance matrix **A** (2) (continued on next page...)
**Figure 1 (…continued)**
and a connectivity matrix **B** (3) describing the spatial relationship among sites (see main text for specific decisions for each dataset). The Hadamard product of these two matrices generates the spatial weighting matrix **W** (4), which is then doubly centred and diagonalised, yielding eigenvectors to be used as spatial variables, represented below by matrix **X**. **Obtaining realistic coefficients for spatial variables (III)**: From a Generalized Linear Model (GLMs) for the relationship between **Y** and **X** (5) we obtained a matrix **C** of realistic regression coefficients (6). **Using non-zero coefficients to model new presence/absence data with pre-defined spatial structure (IV)**: We sampled different numbers of non-zero coefficients from **C** under 14 distinct scenarios (see main text) to build a new matrix **C\*** and then left-multiplied **C\*** by **X** (7) to obtain matrix **Y\***. This matrix represented the logit predicted probabilities of presence or a matrix of log abundances, depending on which of two models that differed, respectively, in assumptions regarding absences as real (simulated presence model, SPM) or artifacts derived from poor sampling (SAM). From Y\* we estimated (8) new presence/absence data **Y⋆** containing the spatial structure defined by **C\***. **Using GLM/AIC and RDA/FW to select spatial models using the simulated presence/absence data (V)**: Finally, we regressed **Y⋆** against **X** using the GLM/AIC and RDA/FW frameworks (9) to assess which MEMs would be correctly selected by those two methods. The performance of each method was mainly assessed by the proportion of MEM variables that were correctly included or excluded from final models by each method (10).

triangulation (**B**) with Euclidian weights (**A**). Despite some degree of connectivity among all sites, pairs of sites could be mostly associated not to their immediate neighbours but rather as a function of their distances. This is due to cultural differences in land management. For example, northern and western islands share cultural histories, which is reflected in species composition (*Lewis, Pakeman & Marrs, 2014*). Directional spatial processes in ecological data, such as those observed in rivers, are well described by a special case of MEMs called asymmetric eigenvector maps (AEM, *Blanchet, Legendre & Borcard, 2008b*), which were used for constructing variables for dataset 3. In MEMs, larger eigenvalues are associated with broader-scale spatial structures while smaller eigenvalues represent fine-scale spatial structures. This allowed us to control the spatial scale of variation in community structure. Dataset 1 had 16 positive MEMs from 42 sites, dataset 2 had 30, and dataset 3 had 12 AEM variables with positive autocorrelation. For computation of the MEMs for the three datasets we used the packages *adespatial* (version 0.3–7, *Dray et al., 2019*) and *spdep* (version 0.7–4, *Bivand & Piras, 2015*; *Bivand, Hauke & Kossowski, 2013*).

## Simulating communities with chosen spatial drivers

We simulated realistic communities with known spatial structure, based on the three datasets. We used spatial eigenvectors as explanatory variables. We varied the number of MEMs with non-zero coefficients and created new binary (presence/absence) communities (with the same number of sites and same expected number of species as the real ones) using two different modelling scenarios. These simulated communities reflected the effect of those MEMs with non-zero coefficients. By varying the number and ordering of the non-zero coefficients, we could therefore control the spatial structure and scale of the simulated community data (see scheme in Fig. 1 and Table 1).

In order to simulate new binary communities under the simulated presence method (SPM, in which species are always detected if present), we first estimated a coefficient matrix **C** of size ($m$ variables + 1 (first) row with intercepts) × $p$ species from each real data set. This was achieved using the manyglm function with binomial errors in R package *mvabund*

**Table 1 Simulation scenarios for the three datasets as described in main text.** Distribution of MEM variables with non-zero coefficient under each simulation scenario in all three datasets (A = marine algae from Ilha Grande Bay, $m = 16$; B = Scotland grasslands, $m = 30$; C = freshwater insects, $m = 12$). Rows and columns define all simulation scenarios regarding the number of variables to be used and their position. Rows represent the number of non-zero variables to be included based on set $K$ (see main text), whereas columns define the scaling of these non-zero variables, i.e. position to which those non-zero variables would be assigned. Scaling 1 assigned non-zero coefficients only to MEMs associated with larger eigenvalues representing broader spatial scales. Scaling 2 assigned non-zero coefficients only to MEMs associated with smaller eigenvalues, representing finer spatial scales. Scaling 3 assigned non-zero coefficients to MEMs representing a range of spatial scales. Cells contain sets of indices of explanatory variables. When nVar=0, none of the variables had non-zero coefficients.

| | | Scaling | | |
|---|---|---|---|---|
| | | **1 (only broad)** | **2 (only fine)** | **3 (mixed)** |
| (A) | 0 | None | – | – |
| | $\lfloor m/6 \rfloor$ | $\{1, 2\}$ | $\{15, 16\}$ | $\{1, 16\}$ |
| | $\lfloor m/3 \rfloor$ | $\{1, 2, 3, 4, 5\}$ | $\{12, 13, 14, 15, 16\}$ | $\{1, 2, 3, 15, 16\}$ |
| | $\lfloor m/2 \rfloor$ | $\{1, 2, \ldots, 8\}$ | $\{9, 11, \ldots, 16\}$ | $\{1, 2, 3, 4, 13, 14, 15, 16\}$ |
| | $\lfloor 3m/4 \rfloor$ | $\{1, 2, \ldots, 12\}$ | $\{5, 7, \ldots, 16\}$ | $\{1, 2, \ldots, 6, 11, 12, \ldots, 16\}$ |
| | $m$ | $\{1, 2, \ldots, 16\}$ | – | – |
| (B) | 0 | None | – | – |
| | $\lfloor m/6 \rfloor$ | $\{1, 2, 3, 4, 5\}$ | $\{26, 27, 28, 29, 30\}$ | $\{1, 2, 3, 29, 30\}$ |
| | $\lfloor m/3 \rfloor$ | $\{1, 2, \ldots, 10\}$ | $\{21, 22, \ldots, 30\}$ | $\{1, 2, \ldots, 10, 21, 22, \ldots, 30\}$ |
| | $\lfloor m/2 \rfloor$ | $\{1, 2, \ldots, 15\}$ | $\{16, 17, \ldots, 30\}$ | $\{1, 2, \ldots, 8, 24, 25, \ldots, 30\}$ |
| | $\lfloor 3m/4 \rfloor$ | $\{1, 2, \ldots, 22\}$ | $\{6, 7, \ldots, 30\}$ | $\{1, 2, \ldots, 11, 21, 22, \ldots, 30\}$ |
| | $m$ | $\{1, 2, \ldots, 30\}$ | – | – |
| (C) | | | | |
| | 0 | None | – | – |
| | $\lfloor m/6 \rfloor$ | $\{1, 2\}$ | $\{11, 12\}$ | $\{1, 12\}$ |
| | $\lfloor m/3 \rfloor$ | $\{1, 2, 3, 4\}$ | $\{9, 10, 11, 12\}$ | $\{1, 2, 11, 12\}$ |
| | $\lfloor m/2 \rfloor$ | $\{1, 2, \ldots, 6\}$ | $\{7, 8, \ldots, 12\}$ | $\{1, 2, 3, 10, 11, 12\}$ |
| | $\lfloor 3m/4 \rfloor$ | $\{1, 2, \ldots, 9\}$ | $\{4, 5, \ldots, 12\}$ | $\{1, 2, 3, 4, 5, 9, 10, 11, 12\}$ |
| | $m$ | $\{1, 2, \ldots, 12\}$ | – | – |

(version 3.11.9, *Wang et al., 2012*), with explanatory matrix $\mathbf{X}$ ($n$ sites $\times$ $m$ positive MEMs + an initial column of 1's). The matrix $\mathbf{C}$ gives the effect of each explanatory variable on the logit-transformed probabilities of presence. The *mvabund* package provides a GLM framework for multivariate response data.

We then created new hypothetical scenarios by generating a new coefficient matrix $\mathbf{C^*}$, of the same size as $\mathbf{C}$, whose elements $c_{kj}^*$ are given by

$$\begin{cases} c_{kj}^* = c_{1j}, \text{if } k = 1, j = 1, 2, \ldots, p, (\text{intercepts}) \\ c_{kj}^* \sim \hat{F}_b, \text{if } k - 1 \in K, j = 1, 2, \ldots, p \\ c_{kj}^* = 0, \text{otherwise}, \end{cases} \tag{1}$$

where $\hat{F}_b$ is the empirical distribution function of $c_{kj}$ ($k = 2, 3, \ldots, m + 1, j = 1, 2, \ldots, p$) (*Evans, Hastings & Peacock, 2000*), and the $b_{kj}^*$ are sampled with replacement. The set $K$ defines to which rows of $\mathbf{C^*}$ the non-zero coefficients were allocated: we studied 14 such sets

(see below and Table 1(A–C)). In other words, we used the originally-estimated intercepts in each simulation (first row of Eq. (1)), and drew those coefficients assigned to non-zero values (second row of Eq. (1)) from the empirical distribution of all the originally-estimated explanatory variable coefficients. We sampled the values of the non-zero coefficients from the empirical distribution in order to simulate plausible but not fixed spatial structures. Table 1 depicts for each dataset how the non-zero coefficients were assigned for each dataset and simulation scenario (see below).

We then calculated predicted probabilities of presence $\hat{p}_{ij}$ for the $j$th species at the $i$th site. Given the matrix $\hat{\mathbf{Y}} = \mathbf{X}\mathbf{C}^*$ ($n$ sites $\times$ $p$ species) of predicted logit probabilities of presence, the predicted probability of presence is

$$\hat{p}_{ij} = \frac{\exp(\hat{y}_{ij})}{1 + \exp(\hat{y}_{ij})}. \tag{2}$$

The simulated presence/absence value for species $j$ at site $i$ was sampled from a Bernoulli distribution with success probability $\hat{p}_{ij}$. The result is a community matrix with the same number of sites and the same expected number of species as the real community, and with realistic coefficients for spatial eigenvectors. As in the maximum likelihood estimation done by manyglm (*Wang et al., 2012*), species and sites were assumed conditionally independent when generating simulated presence/absence data, given the values of the explanatory variables. Our simulated communities correspond to the simple case in which presence/absence patterns are affected by environmental variables but not interspecific interactions. Nevertheless, interspecific interactions could well be relevant to real world systems and other models (*Godsoe & Harmon, 2012*; *Anderson, 2017*).

Since GLMs are specified correctly for presence/absence data generated this way, we would expect them to perform well. We therefore devised a second ecologically meaningful simulation method in which absences arise from the sampling protocol, called the simulated abundance method (SAM). The two simulation methods differ in whether they assume we have true absences or sampling-related absences. Note that it is not possible to simulate binary data directly using RDA, because RDA does not generate predicted probabilities of presence. Instead, we treated $\hat{\mathbf{Y}}$ as log expected abundances and exponentiated each element to get expected abundances $\lambda$. Then we calculated the probability of detecting the species under Poisson sampling (i.e., the probability of drawing a value of at least 1 from a Poisson distribution with parameter $\lambda$), which is

$$\hat{p}_{ij} = 1 - e^{-\lambda}. \tag{3}$$

Finally, we generated a Bernoulli random variable with success probability $\hat{p}_{ij}$ to produce a simulated presence-absence observation. Both GLM and RDA are mis-specified for data generated in this way. Codes for both the SPM and SAM simulation frameworks and all the datasets used in our simulations are available as supplemental information (Data S1–Data S3).

We compared GLM and RDA variable selection under up to 14 different scenarios, differing in the number of non-zero coefficients (*nVar*) and whether these coefficients were associated with fine or broad spatial scales. We simulated up to six different choices of

the number of MEM variables creating the spatial structure in the data (i.e. having non-zero coefficients): none, approximately one sixth, approximately one third, approximately half, approximately three-quarters, and all (Table 1(A–C), rows). We also simulated three different spatial scales of the patterns. As mentioned above, MEMs associated with larger eigenvalues represent broader spatial scales. We ordered the MEMs in descending order of eigenvalues and arranged the non-zero coefficients within matrix $C^*$ in three different ways (Table 1 (A–C), columns): only broad-scale MEMs with non-zero coefficients (scaling 1); only fine-scale MEMs with non-zero coefficients (scaling 2); half broad-scale, half fine-scale (scaling 3). Because not every combination of number of non-zero coefficients and spatial scaling is possible (e.g., it is not possible to assign one non-zero coefficient in scaling 3), there were 14 possible combinations overall for each dataset (Table 1). The main steps of the simulation scheme are summarized in Fig. 1.

## RDA and GLM

We used the default RDA function from the R package *vegan* (version 2.5-6, *Oksanen et al., 2019*), with simulated community composition as the response variable, and MEMs associated with positive eigenvalues generated from geographical coordinates of the sample sites as explanatory variables. In order to perform a transformation-based RDA (*Borcard, Gillet & Legendre, 2011*; *Blanchet et al., 2014*)  we used the Ochiai coefficient, which is the Hellinger transformation analogue for binary data, as recommended by *Legendre & Gallagher (2001)* and *Borcard, Gillet & Legendre (2011)*.

Binomial GLMs were fitted to the same data using the manyglm function in R package *mvabund* (*Wang et al., 2012*). We fitted our models using a logistic regression (logit link function for binomial response), with species compositional data as the multivariate response variable and MEMs as predictors. No interaction terms were included, following common practice in spatial modelling of community data.

## Comparing model selection between RDA and GLM frameworks

We compared the results of model selection between the approach usually taken in the RDA and a somewhat-similar approach for GLMs. For RDA, we used the forward selection with double stopping criterion following (*Bauman et al., 2018a*; *Bauman et al., 2018b*), beginning with a global test of significance (model with all spatial predictors) and carrying on with the variable selection if the global model was significant. The forward selection itself consists of a stepwise procedure including in the model the variable contributing the most to the adjusted $R^2$. The procedure stops either when the next variable with the highest contribution is not significant (first stopping criterion) or causes the adjusted $R^2$ to be bigger than that of the global model (i.e., containing all variables; second criterion). This is implemented in the function ordiR2step in the *vegan* package (*Oksanen et al., 2019*). For GLM, we used forward selection with a stopping rule based on minimum Akaike Information Criterion (AIC) (*Akaike, 1973*; *Wagenmakers & Farrell, 2004*). The selection procedure started from a model with intercept only and added one explanatory variable at a time, until no further improvement in the sum of AIC over each of the response variables was possible. We used this approach because the usually large number of MEMs makes it difficult to compare the AIC sum over all possible GLMs.

The performance of each method on simulated data was mainly assessed by two criteria. First, we assessed how many MEMs with zero coefficients were incorrectly included in the final model. Second, we assessed how many MEMs with non-zero coefficients were incorrectly excluded from the final model. Also, we assessed overall accuracy (score) as the percentage of MEMs whose inclusion/exclusion status was correct. The goals of ecological studies are usually not directly related to the inclusion/exclusion of individual MEM variables, but instead to identify spatial pattern, represented by a linear combination of MEMs. However, since the MEMs form a basis for the space spanned by the transformed spatial weighting matrix, such a linear combination is unique (*Fraleigh & Beauregard, 1995*, pages 197-198). Furthermore, the MEMs are orthogonal, so that each represents a qualitatively distinct aspect of spatial pattern. Therefore, if an individual MEM is incorrectly included or excluded, the estimated spatial pattern is qualitatively wrong.

We further explored the ability of each method to capture spatial pattern using a graphical approach (Article S1). For each real dataset and each method, we haphazardly picked one simulated data set. We plotted the MEM decompositions of both the true and estimated spatial patterns. We chose the scenarios in which each method had the worst performance in terms of correctly including/excluding variables, in order to determine whether in such cases, overall spatial pattern would still be captured.

Finally, we calculated how much of the variation in response variables was explained by each method using the adjusted $R^2$ for the linear model in RDA and its analogue for GLMs, the $D$-value (*Tjur, 2009*). These two values cannot be directly compared since they are not exactly equivalent, but their results could yield interesting insights and are made available as supplemental information (see table results in Data S4).

For each of the combinations of conditions in Table 1, 1,000 simulated data sets were generated under each of SPM and SAM. For each simulated data set, spatial explanatory variables were selected using both GLM/AIC and RDA/FW.

## RESULTS

Overall, GLM/AIC outperformed RDA/FW in selecting spatial explanatory variables when data were simulated under either SPM or SAM in all three scaling patterns (Fig. 2).

In general, GLM/AIC had fairly predictable performance: it performed nearly perfectly when few or none of the available variables had non-zero true coefficients (i.e., $nVar = 0$, $[m/6]$, $[m/3]$ or $[m/2]$), but was less accurate when many or all the variables had non-zero true coefficients ($nVar = [3m/4]$ or $nVar = m$) (blue lines in Figs. 2A–2E). There was also some discernible pattern in RDA/FW's scores: it performed best at $nVar = 0$ and $nVar = m$, with intermediate values showing a considerable decrease in selection success. The loss of accuracy for intermediate values of $nVar$ (drop in red lines across different $nVar$ values in Fig. 2 A-E) varied substantially among datasets, making general inferences about results more difficult. There was little difference between the results from the SPM (Figs. 2A, 2C, 2E) and SAM simulations (Figs. 2B, 2D, 2F).

It is also noteworthy that when the model had a smaller number of variables to select from (River dataset 3 with 12 MEMs), scores in GLM/AIC were higher, with virtually no

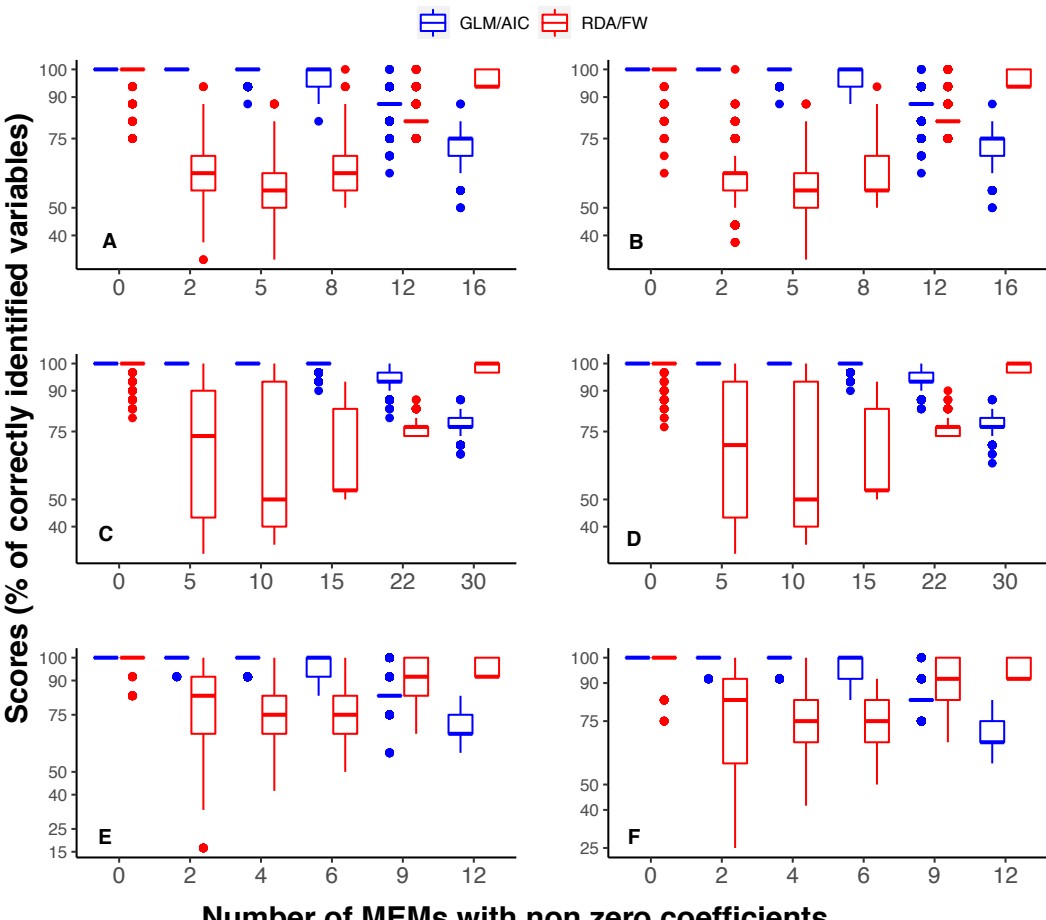

**Figure 2 Overall performance comparison between GLM/AIC (blue) and RDA/FW (red) methods on simulated presence/absence data.** Scores were measured by counting the percentage of MEMs correctly included/excluded from the final model out of the total number of variables in each dataset (1 = 16, 2 = 30, 3 = 2). This comparison was made across varying numbers of MEMs with non-zero coefficients (*x* axis). (A, B) simulated data based on subtidal macroalgae in Ilha Grande Bay; (C, D) data based on plant species from Scottish grassland and (E, F) data based on aquatic macroinvertebrate insect species from a river in Brazil. A, C and E depict results where community presence/absence data was simulated directly from real coefficients (SPM, see main text) whereas B, D and F show simulation results where presence/absence data was estimated from expected abundances (SAM).

incorrect inclusion of variables, and incorrect exclusion of variables occurring on average in only approximately 6% of all 14000 simulations over the whole set of replicates (Fig. 3E). Under the same conditions, RDA/FW's rate of success was approximately 81%, incorrectly including variables at a rate of 18% (incorrect exclusions represented less than 1%) as depicted in Fig. 3E.

Under both the SPM and SAM simulation methods, GLM/AIC differed substantially from the RDA/FW framework in regard to the type of errors it most often produced. GLM/AIC had virtually no incorrect inclusion of variables (Fig. 3, blue). However, when $nVar = [3m/4]$ or $nVar = m$ some variables that should be included in the final model

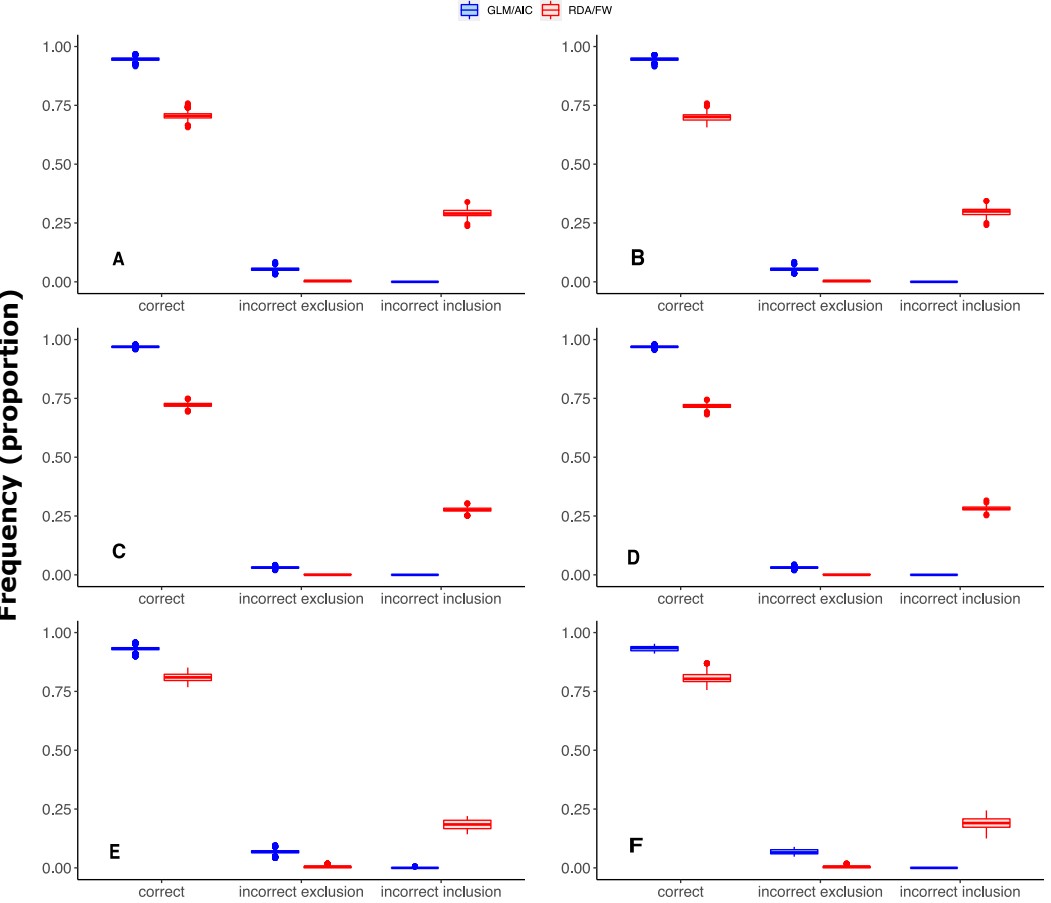

**Figure 3** **Differences in performance between GLM/AIC and RDA/FW frameworks regarding the proportion of incorrect inclusions/exclusions of explanatory variables across 1,000 simulations for each method.** Panels A, C and E depict results where community presence/absence data was simulated directly from real coefficients (SPM, see main text) whereas B, D and F show simulation results where presence/absence data was estimated from expected abundances (SAM). Panels A and B depict results for simulated data based on subtidal macroalgae in Ilha Grande Bay; C and D represent data based on plant species from Scottish grassland; and E and F represent data based on aquatic macroinvertebrate insect species from a river in Brazil. Darker lines represent mean values.

were left out. Nevertheless, GLM/AIC never had less than around 90% accuracy over all three datasets (overall mean = 96 ± 1.3% against 71 ± 1.7% from RDA/FW). On the other hand, RDA/FW often included more variables than it should in the model (Fig. 3, red). Such errors especially occurred when $0 < nVar \leq [3m/4]$. Under some conditions, up to one third of the variables selected by RDA/FW had zero coefficients.

MEM decompositions of true and estimated spatial structure provided a visual assessment of the extent of the misspecification yielded by each method (Article S1). In all three datasets, the worst performance of GLM/AIC corresponded to those models in which it should have included all MEM variables (Fig. 2). Those scenarios represented communities structured at all spatial scales (broad, intermediate and fine). Despite

incorrectly excluding several individual variables, GLM/AIC was capable of selecting subsets of variables that corresponded to all those scaling categories ( Articles S1.2–S1.7). In contrast, RDA/FW performed worse when there were few spatial variables (nVar = 5, nVar = 10 and nVar = 2 for datasets 1, 2 and 3, respectively). Under those conditions, incorrect inclusion of variables also resulted in the inclusion of incorrect spatial scales. For example, in one simulation from dataset 1 (Article S1.8) the true spatial structure contained only five MEMs describing finer spatial scale patterns (scaling 2 = MEMs 12-16). However, the final model selected by RDA/FW included 13 variables describing both broad (MEMs 1-6) and intermediate spatial scales (MEMs 9, 11), along with the correct ones (Article S1.9). Similar results were found in all three datasets (Articles S1.10–S1.13). Moreover, these incorrect inclusions of individual variables by RDA/FW resulted in the inclusion of MEM variables associated with eigenvalues substantially different from the correct ones, representing spatial scales much larger than those actually present in the data (Article S1.14). For matters of space, we only plotted one failure example from each dataset for both GLM/AIC and RDA/FW. However, the correct spatial structures within simulated communities and those structures retrieved by both methods in all our simulations scenarios are available as supplemental data (Data S5).

Under SPM simulations, the scale of spatial pattern (fine, broad or mixed: scaling 1, 2 and 3, respectively) had negligible effect on GLM/AIC performance (Figs. 4A, 4C, 4E). A slight difference in variable selection scores between scaling 1 to 2 and 3 was only found in one modelling condition (Fig. 4, $nVar = [3m/4]$). On the other hand, scaling often affected the performance of RDA/ FW, although there was no obvious general pattern across different conditions and datasets (Figs. 4A, 4C, 4E). Under SAM simulations, both frameworks performed similarly to what was observed under SPM (Figs. 4B, 4D, 4F).

## DISCUSSION

Here, we showed that a GLM/AIC-based method for finding spatial structure in communities outperformed an RDA/FW-based method, for presence-absence data simulated under two different ecologically plausible scenarios about how absences arise. We based our simulated datasets on real datasets from marine, terrestrial and freshwater data. Notably, differences in assumptions about how absences arise made little difference to performance. This might be due to the structure of our community presence/absence datasets, which (like most ecological datasets) had many rare species and, therefore, many expected abundances close to zero. In such cases, the relationship between the community data and explanatory variables could be approximated by a binomial GLM with a logit link function, even if this was not the correct model (as in the SAM simulations). We therefore focus below on general patterns that apply equally to both assumptions about absences, rather than on the details of these assumptions.

In selecting spatial explanatory variables, GLM followed by AIC-based model selection (GLM/AIC) performed better than the widely-used approach of RDA followed by forward selection (RDA/FW). Not only did GLM/AIC have better performance overall, but its performance varied little between simulation conditions (Fig. 2). In contrast, RDA/FW performed unpredictably, but often retained too many explanatory variables (Fig. 3).

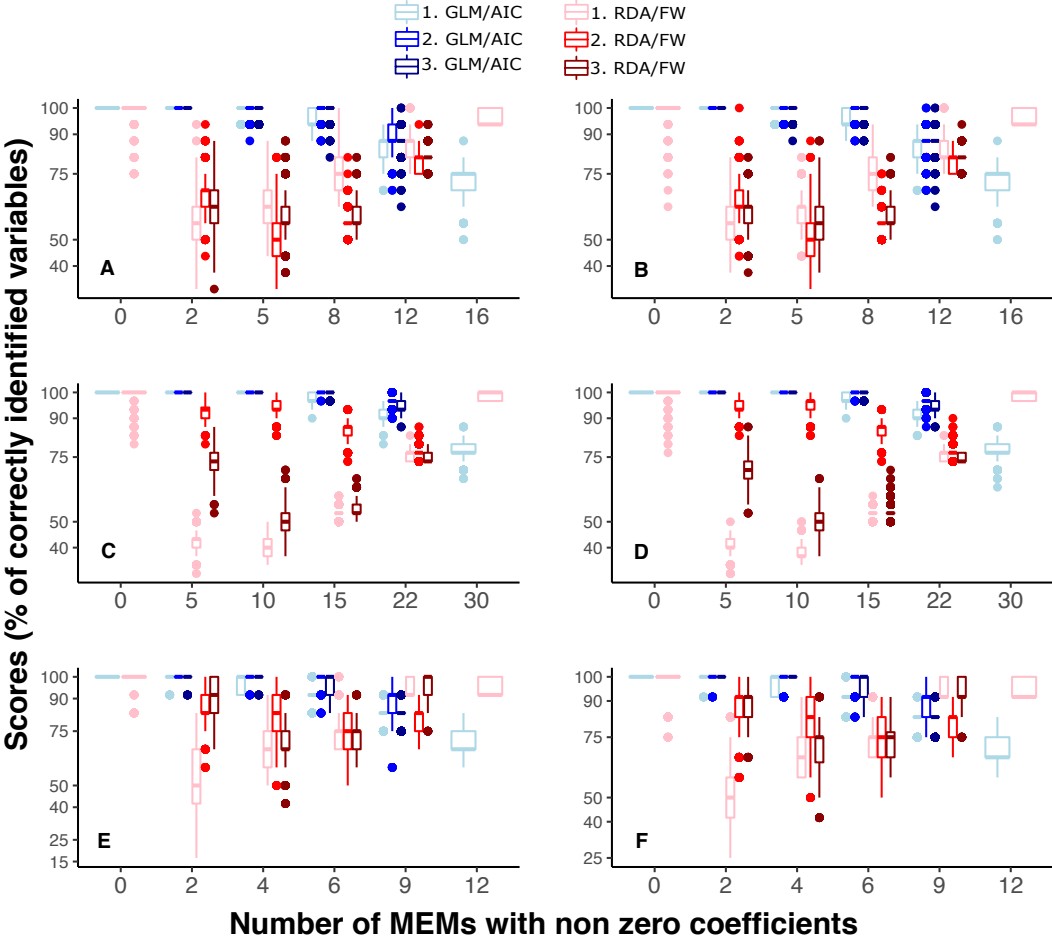

**Figure 4** **Performance of GLM/AIC (blue) and RDA/FW (red) modelling approaches under variation in spatial scales of MEMs with non-zero coefficients.** Spatial scale was defined as broad (1), fine (2) or mixed (3) (where applicable). (A, B) simulated data based on macroalgae in Ilha Grande Bay; (C, D) data based on plant species from Scottish grassland and (E, F) data based on aquatic macroinvertebrate insect species from a river in Brazil. (A, C and E) depict results where community presence/absence data was simulated directly from real coefficients (SPM) whereas (B, D and F) show simulation results where presence/absence data was estimated from expected abundances (SAM, see main text).

The problems arising from data with non-Gaussian error distributions, such as classic community presence and absence data, in a linear modelling framework are not new to science (*Legendre & Gallagher, 2001*; *McCullagh & Nelder, 1989*; *Wolda, 1981*). Classical linear models such as RDA (*Legendre & Anderson, 1999*; *Legendre & Legendre, 2012*) make assumptions regarding constancy of variance in the data (*Ter Braak & Prentice, 1988*) that cannot be true for presence-absence data, even after data transformation (*O'Hara & Kotze, 2010*; *Warton, 2018*; *Warton, Wright & Wang, 2012*). The problem may be negligible in some hypothesis testing situations (*Ives, 2015*). Regardless, incorrectly assuming linearity (and constant variance) may lead to serious problems. Unfortunately, RDA is an algorithmic method that makes implicit decisions about the distribution of variances (*Ter Braak & Prentice, 1988*; *Warton, Wright & Wang, 2012*) and does not

provide the flexibility to separate systematic variation from random variation in the way that statistical models such as GLMs do (*Warton et al., 2015*; and see *O'Neil & Schutt, 2013*) for differences between algorithms and statistical models). New frameworks, such as using GLMs with spatially-structured random effects (followed by variation partitioning to find environmental and spatial components) have also been specifically proposed as a model-based alternative to MEMs (*Ovaskainen et al., 2017*). Despite recent advances showing that better estimates could be obtained by using sensible selection procedures, manipulating the data appropriately and/or by splitting the analysis of the response data over shorter spatial/environmental gradients (*Bauman et al., 2018a*; *Ives, 2015*; *Vieira et al., 2019*), employing statistical models that match the distribution of the response data is better practice in most cases (*Ferrier et al., 2007*; *Warton, 2018*; *Warton et al., 2015*).

Another relevant aspect of the general performances of the two methods concerns the peaks of performance in detecting spatial structure. The scores in the GLM/AIC framework were close to ideal across datasets when the number of variables that should be selected was none or was small relative to the number of variables available. The performance only decayed when many or all of the available variables should have been retained in the final model. Thus, if a few variables are responsible for most of the spatial structure in community composition, GLM/AIC will usually outperform RDA/FW (Fig. 2). Considering that the majority of effects could be derived from a small number of causes (*Sullivan, 2019*) in many biological systems, GLM/AIC could presumably perform well on many real systems. On the other hand, RDA/FW worked best precisely in situations thought unlikely in real systems, when no spatial structure is present among communities (where GLM/AIC also performed equally well), or when composition is structured at all possible spatial scales (i.e., $nVar = 0$ and $nVar = m$, respectively). Moreover, when the model had a small number of variables to select from (River dataset, Figs. 3E–3F), performance of RDA/FW was very variable.

The two approaches also differed in the ways they failed. GLM/AIC more often included too few variables, while RDA/FW more often included too many. This was consistent among all three datasets under SPM and SAM simulations (Fig. 3) and is in contrast with results from previous studies where GLMs produced higher Type I error rates compared to a linear model (*Ives, 2015*). For beta diversity studies, where the aim is to identify the most important variables associated with differences in community composition, leaving out a few variables that affect composition is better, in our opinion, than including many variables whose effects are not important. On the contrary, in other scenarios such as when one tries to select pivotal attributes that could be important for the conservation of a population or community, it might be better to accept a higher risk of including spurious variables. Furthermore, model selection problems involve a trade-off between bias and variance, with inclusion of unnecessary variables inflating the uncertainty in parameter estimates (*Miller, 1990*). Using AIC is often a good way to deal with this trade-off (*Anderson, Burnham & Thompson, 2000*), and in our simulations, an AIC-based approach worked well. Thus, we suggest that GLM/AIC will usually outperform RDA/FW in selecting spatial explanatory variables for presence/absence community composition data. Unfortunately, AIC-like statistics are not recommended for constrained ordination methods such as RDA, and therefore its use cannot be trusted (see below and *Bauman et al.,*

*2018a* for details). When different RDA-based procedures were systematically compared, the commonly (mis)used combination of RDA and AIC model selection produced the worst results, yielding inflated Type I errors rates (*Bauman et al., 2018a*). Therefore, the benefits from AIC in dealing with the bias and variance trade-off do not apply to RDA or related ordination methods. Despite our interest in some attributes of the MEMs for our simulations, such as differences in model performance under varying spatial scales, we hypothesize that the results demonstrated here hold true for other types of explanatory variables (e.g., environmental) not tested here.

The spatial scale represented by the MEMs had a negligible effect on GLM/AIC's performance, with only one condition in one dataset slightly differing in results between different scales (see Fig. 4 when the number of non-zeros is $\lfloor 3m/4 \rfloor$). In contrast, RDA/FW's performance was strongly affected by spatial scale (Fig. 4). In real systems, where the spatial scale at which community composition varies is not known *a priori*, the performance of RDA/FW could therefore be unpredictable. The uncertainty around RDA/FW performance over differing spatial scales could be especially troublesome for analyses involving processes that may not be constant along spatial/environmental gradients, as commonly observed for rates of species turnover, for example (*Ferrier et al., 2007*; *Fitzpatrick et al., 2013*).

## CONCLUSIONS

We discourage the use of traditional RDA/FW to search for spatial descriptors of variation in multivariate presence/absence data sets of moderate size, although large datasets could potentially overcome the issues found here. Instead, we recommend the GLM/AIC framework, in which the relationship between the response and its predictors is modelled in a way that respects the nature of the response. Similar recommendations are likely to apply to other forms of community abundance data with non-normal error distributions (e.g., count data with many zeros or proportional data, *Bolker et al., 2009*; *Warton, Wright & Wang, 2012*; *Warton et al., 2016*).

## ACKNOWLEDGEMENTS

We thank the James Hutton Institute, Aberdeen, for providing data. We are also grateful for Dr Petr Šmilauer for valuable suggestions given at BES 2015 and Dr Ian Smith for technical support.

### Funding

This work was supported by National Council for Technological and Scientific Development, (CNPq) within Science without Borders scholarship scheme (Lélis Carlos-Júnior) and CNPq-305330/2010-1 (Joel Creed). Additional funding was provided by Brazilian Coordination for the Improvement of Higher Education Personnel (CAPES); CAPES- Ciências do Mar (Joel Creed 1137/2010); Fundação Carlos Chagas Filho de Amparo à Pesquisa do Estado do Rio de Janeiro (Joel Creed., FAPERJ-E-26/111.574/2014

and E26/201.286/2014). Finally, Rob Lewis received funding from EU's Marie Skłodowska-Curie action (Grant 703258) and Rafael Feijó-Lima received funding from CAPES program Science Without Borders –project n. 166/2012. The funders had no role in study design, data collection and analysis, decision to publish, or preparation of the manuscript.

## Grant Disclosures

The following grant information was disclosed by the authors:
National Council for Technological and Scientific Development, (CNPq).
Science without Borders scholarship scheme (Lélis Carlos-Júnior).
CNPq: 305330/2010-1.
Brazilian Coordination for the Improvement of Higher Education Personnel (CAPES).
CAPES- Ciências do Mar:  1137/2010, FAPERJ-E-26/111.574/2014, E26/201.286/2014.
EU's Marie Skłodowska-Curie action: 703258.
CAPES program Science Without Borders –project: 166/2012.

## Competing Interests

The authors declare there are no competing interests.

## Author Contributions

- Lélis A. Carlos-Júnior conceived and designed the experiments, analyzed the data, prepared figures and/or tables, authored or reviewed drafts of the paper, and approved the final draft.
- Joel C. Creed conceived and designed the experiments, authored or reviewed drafts of the paper, and approved the final draft.
- Rob Marrs, Rob J. Lewis and Timothy P. Moulton analyzed the data, authored or reviewed drafts of the paper, and approved the final draft.
- Rafael Feijó-Lima , analyzed the data, prepared figures and/or tables, authored or reviewed drafts of the paper, and approved the final draft.
- Matthew Spencer conceived and designed the experiments, performed the experiments, analyzed the data, authored or reviewed drafts of the paper, and approved the final draft.

## Field Study Permissions

The following information was supplied relating to field study approvals (i.e., approving body and any reference numbers):

The algae presence/absence data was collected under permission from IBAMA/RJ (031/04-IBAMA/RJ). The freshwater macroinvertebrate data was collected under the permit from INEA-RJ (019-2014).

## Data Availability

The R script code and files needed to perform all simulation frameworks and the results are available in the Supplementary Files.

## Supplemental Information

Supplemental information for this article can be found online at http://dx.doi.org/10.7717/peerj.9777#supplemental-information.

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
