# Peer review of "Generalized Linear Models outperform commonly used canonical analysis in estimating spatial structure of presence/absence data"

_PeerJ, doi:10.7717/peerj.9777_

## Round 0.1 · original submission · Major Revisions

Thank you for your submission and your patience in this process. I have received three detailed reviews for this manuscript. Across the three, I am inclined to request a Major Revision, despite a reviewer noting rejection. Please look carefully at each. Please note that the rejection review points out a MAJOR ERROR in identifying a Bug in your R code and noting your use of AIC is incorrect. It is also pointed out that variable selection is incorrect. This too needs to be addressed.

Please start with a detailed review and overhaul of the paper following the rejection review. It is an excellent road map to improvement. Without addressing the error noted in the code, this paper cannot be considered for publication.

There are two additional detailed Major Revision reviews and both provide clear feedback on needs to revise the manuscript.

I look forward to seeing the results when errors have been fixed and appropriate selection techniques used.

·

Basic reporting

This paper aims to compare canonical analysis (RDA on Hellinger-transformed data) to GLMs based approaches and shows that the second outperforms the first in estimating spatial structure of presence/absence data. The paper is quite well written but I think that the reference to Ives (2015) could be considered in the introduction about data transformation.

AIC for RDA has no statistical basis (this is stated several times in the help pages of the vegan package). I would suggest to use only forward selection based on adjusted R2. Moreover, the authors should consider the recent works on selection that shows that a global test should be perform prior to forward selection to obtain correct level of type I error (Bauman, D., Drouet, T., Dray, S., & Vleminckx, J. (2018). Disentangling good from bad practices in the selection of spatial or phylogenetic eigenvectors. Ecography.). Authors should also consider MEM instead of PCNM as the latter are less powerfull (see Bauman, D., Drouet, T., Fortin, M. J., & Dray, S. (2018). Optimizing the choice of a spatial weighting matrix in eigenvector‐based methods. Ecology, 99(10), 2159-2166.)

Authors provides data and R scripts. Thanks !

Figures and caption could be improved. I cannot understand what is the scores for figure 2 as it counts both the numbers of correctly excluded/included variables. What is the unit? How it should be interpreted for intermediate levels? Figures would be easir to read by using a 2 x 3 layout so that GLM and RDA models are on separate columns. Text on figures should be improved (Number of PCNM.. on figure 2, error I/II and correct have no meanings -> type I error, etc)

Experimental design

I think that the title/message does not reflect what is done is the work for several reasons:
- in the paper, the authors compare variable selection procedures, not RDA vs GLM methods per se. RDA is an ordination technique (i.e., dimension reduction) and do not include a selection procedure. Hence, I think that main message should talk about selection procedure, not about GLM/RDA method. The main outputs of RDA are biplot/triplot to summarise the structures of ecological communities and these aspects are really not considered in the paper.
- the authors evaluate which variables are included in the model compared to a "true model", not the spatial structure obtained (as stated in the title). The predicted structure could be similar to the truth even if not the same explanatory variables are included in the model. More generally, I think that these methods (PCNM, MEM, etc.) aims to identify spatial structures as a combination of spatial predictors. Taking each spatial predictor separately makes no real sense for ecologists, what is important is the linear combination that results in predicted spatial pattern. Hence, I think that the spatial case makes little sense for such comparison and I would suggest to use true environmental descriptors so that the study would make more sense.

I think that authors should better use an ecological model to simulate data than GLM/RDA-based. I found the RDA-based model very difficult to justify in an ecological viewpoint and the GLM based is too close to the model tested. Moreover, 0/1 data are often the results of a sampling protocol not of species responses. Hence, it would make much more sense to simulate abundance (e.g., using a gaussian response model) and then simulate a sampling protocol that leads to 0/1 data if the aim of the paper is to focus on presence/absence data. Hence, the comparison of methods would be fair as data are simulated under an independent model.

The protocol should be better described. They suggest that they used the forward.sel function but in fact they use ordiR2step. Moreover I am not sure that they used the procedure of Blanchet et al (2008) that contains a global test prior to forward selection (this is not done in the code provided). I think that the GLM model (implemented in manyglm) should be better described in a mathematical viewpoint (which terms/interactions are considered). Please describe methods properly not only by giving a name of a function. Also, provide version number of packages to ensure the reproducibility of analyses

I think that the part focusing on broad, medium and fine scales is not particularly relevant. What are the hypotheses ? Did you expect that some scales are better described than other ? Why? Does your finding relevant for non-spatial variables? I would suggest to remove this part.

Validity of the findings

The most important problem is that I found a bug in the code provided by the author. In the rda.func/rda.func2 functions, the authors did not identify correctly which variables are selected in the model. They take 'rda.selec$CCA$QR$pivot' as the indexes of the variables selected, and this is definitely not the case. This induces that all results provided are wrong and could not be discussed. Hence, I will not produce a detailed review of the rest of the paper as most results would probably change if this bug is corrected.

Conclusions about the generality of findings seems exaggerated. Results obtained concerns only some particular size of data tables, with only 0/1 data and spatial predictors. Please be more cautious.

Additional comments

Please correct the bug and rerun simulations to see how results changes. State clearly that your paper focus on comparison of selection procedures, not on GLM vs RDA outputs. If you focus on spatial patterns, I think that what is relevant is the linear combination of spatial variables, not the identity of spatial variables selected (these variables are artificial and have no biological meaning). Thus I suggest to focus on the precision of the predicted spatial patterns (see Bauman et al, 2018, Ecography). If you do not agree, explain why it is important to know which PCNM are selected and why this is relevant for ecological studies.

Reviewer 2 ·

Basic reporting

Overall, I find the English language clear and technically correct throughout the manuscript. There were some typos and mistakes:
- l.98: “PCNMS” instead of “PCNMs”.
- Legend Figure 2: “coefficients” instead of “coefficents” in the x-label; part of the first sentence is missing, indeed not all simulations were made under GLM.
I think the authors made several errors regarding which subgraphs represent which datasets. Shouldn’t it be (A,B) dataset 1, (C,D) dataset 2, etc ?
- Legend Figure 3: I think the authors made several errors regarding which subgraphs represent which datasets. Shouldn’t it be (A,B) dataset 1, (C,D) dataset 2, etc ?
- Legend Figure 1: “then” instead of “them”; the dot after one of the B is written in bold which should not be; one of the B*is not in bold when it should be (for consistency); the sentence “Matrix B was them manipulated to create B* in which non-zero coefficients drawn from the original B.” does not seem complete;
- Legend Table 1: “A)” should be lowercase for consistency
- l.423: “Willey” instead of “Wiley”

I think the language could be improved in some places to make the text, figures and equations more easily understandable for an international audience:
- l.67: I suggest replacing “axes” by “variables” for clarity.
- l.98-100: The last sentence of the paragraph will not be clear for all readers. You should explain why you did not generate under RDA.
- l.119-120: The authors calculated the PCNMs/MEMs from the geographical coordinates of the sample sites, this latter information could be mentioned here for added clarity and as a link to the datasets.
- l.136-142: 1) At this point of the manuscript, it will not be very clear for everyone in the audience if “non-zero coefficients” are kept/replaced by 0, depending on the set of PCNM that were chosen, or if they are systematically resampled across PCNM in the new B* matrices (which is what the authors have done). I would suggest providing an explicit B* example matrix in Fig.1, e.g. with the last dataset as it is the simplest.
- l.144: It would be good to indicate that the first row is the one consisting of the intercepts.
- l.153: It could maybe be easier to understand for some in the audience if you added “(intercepts)” next to the first line of the equation.
- l.240-241: The authors described thirteen scenarios. Not three. I suggest they use “three scaling patterns” for consistency.
- Figure 2: X which is the matrix of explanatory variables is not appropriately described in (II), indeed it should represent PCNMs in columns with each PCNM describing one value per site (not per species as is currently indicated).

I found no problem with the article structure although some additional references to the figures could be added IMHO:
- l.119-120: I believe (Fig.1) should be added here.
- l.136-142: I believe (Fig.1) should be added here.

I found that the manuscript generally provided decent context/background but I found some potential gaps that could easily be filled:
- l.84-87: The first and last references focused on count data and demonstrated the consequences of not properly specifying the mean-variance relationship for count data, not for presence/absence data. This does raise questions about the consequences of mis-specifying this relationship for presence/absence data, but there is yet to be an evaluation of the consequences. I think this should be made clearer and would improve the manuscript.
- l.87-88: Data transformations indeed do not solve the issue of respecting mean-variance relationships, although they have been used/useful(?) to create widely used dissimilarity coefficients. Maybe this could be made clearer earlier in the paragraph or in the same sentence. One could also cite (Legendre & Gallagher 2001). Warton et al 2012 seems like a relevant reference to add here, as they dealt directly with this for count data.
- l.88-89: I believe Wang et al (2010) (mvabund package) and Yee (2006) (constrained additive ordination) may be earlier relevant references here.
- l.94-95: I believe one could list them for count data (e.g. Warton…) as there are few such studies and highlight that it has not been done for presence/absence.
- l.150-161: I think a reference with a similar approach regarding the reallocation of coefficients to create realistic simulations would be welcome.

Experimental design

The research is original, inspired by recent advances in multivariate community modelling which seek to improve our ability to respect complex mean-variance relationships in model-based approaches. This manuscript fits within the Aims and Scope of the journal. The question is defined clearly and results are expected to fill an important gap.

I have a few methodological concerns, however:
- l.125-128: Is negative spatial auto-correlation always neglectable? If not, I believe there needs to be a justification for not including it.
- l.136-142: It is not clear which test was applied by the authors to define was is a non-zero coefficient and what is not.
- l.172: “species and sites were assumed conditionally independent”, this assumption may raise serious questions about the validity of the simulations. Isn’t this assumption and the subsequent mixing of coefficients making all trophic relationships (e.g. dataset C) and competition (e.g. datasets A and B) irrelevant to the spatial distribution of species in the communities considered in the three datasets? We know species interactions can shape spatial variation within communities (e.g. van Damm 2009, Bascompte 2009, Meier et al 2010, Pelissier et al 2010 …). I think justification of why species interactions and the associated natural spatial covariance can be disregarded in the making of realistic simulations, is expected and should be given by the authors.
- l.188-190: The authors should provide a justification for dividing the number of PCNMs to assign coefficients to, the way they did. In other words, why did the authors focused on very low numbers (0, 1, 2) rather than have equally spaced numbers (i.e. 0, one-sixth, one third, half, two thirds, five-sixths, max)? One possible issue I see is that if GLM would be better than RDA for low numbers of spatial predictors (and vice versa), then their choice for choosing numbers of PCNMs assigned with non-zero coefficients would necessarily favour GLM for a casual reader of this manuscript.

Validity of the findings

Data seems robust and analyses to have produced sound results which support conclusions which are in turn well connected to the original question and hypothesis.

I have however several comments regarding the findings

- l.176-185: I find that the authors addressed the issue of not being able to simulate binary data directly under RDA clearly and cleverly. However, the fact that GLM can be simulated under directly whereas it is not possible for RDA, could be a weakness as this discrepancy necessarily favour GLM. While their approach seems statistically sound to me, it is hard to know whether the approach they used to simulate under RDA really reflect the actual performance of RDA. This worry is heightened by the fact that all three frameworks had similar error rates when applied to simulations based on RDA-derived coefficients.
- l.240-263: I find Figure 2 and its use in the Results improvable. First, it is unclear whether the authors mixed results from analyses with the three different scaling patterns to make Fig. 2. If that is indeed the case, then it really should be indicated for clarity. Second, the authors describe Fig.2 as representing the rate at which PCNMs are correctly included/excluded in the thirteen different scenarios (seven for the last dataset). However, it seems to me that Fig.2 only addresses the number of PCNMs assigned with non-zero coefficients which were included in the final models. Nothing about PCNMs that were excluded, although the two quantities are obviously not independent.

Additional comments

Your discussion would benefit from more detail. For example, I suggest that you improve the discussion about: a) the results from the RDA-based simulations which have been overlooked in my humble opinion, b) what is left to improve and test in spatial community modelling, c) why you deemed unlikely situations when composition is structured at all or none possible spatial scales (where RDA outperformed GLM in your study).

I think that simulations that would have been independent of the methods you used to select the best spatial predictors, and that would have preserved spatial covariance between species, would have greatly benefited this manuscript.

Reviewer 3 ·

Basic reporting

This is a nice work, in general. The subject of the study is interesting and aims at filling a methodological gap in Ecology, that is, defining the most adapted way to analyze presence/absence data in a spatially explicit manner, using either constrained ordination analyses or Generalized Linear Models.

Overall, the experimental strategy is interesting and promising. However, to me, the manuscript in its current state raises some major concerns that I detail below. Among them, the background and state-of-the-art (both in the Introduction and in the Discussion) need to be improved. In addition, the authors used an old way of performing the spatial analyses, which may compromise some results, or at least prevent responding to the questions of the study in the most accurate manner. Notably, choosing PCNM instead of the up-to-date practice (since 2006) which is to use the more flexible and powerful MEM, is a fundamental problem for a study aiming at recommending good practices in terms of spatial analyses when dealing with presence/absence data. While some studies are still published on the basis of PCNM, this has clearly been shown to be a poor choice in many instances (lack of mathematical formalism, lack of statistical power and accuracy in different common situations, as irregular sampling designs, etc, e.g. Dray et al. 2006, Legendre and Legendre 2012, Bauman et al. 2018b), and if the present work aims at giving recommendations for future studies, it needs to be based on the up-to-date and optimized way of performing the concerned spatial analyses. This would also include discussing the choice of the spatial weighting matrix used to generate the spatial eigenvectors (a key step influencing both the overall power and R-squared estimation accuracy, see Bauman et al. 2018b).

The methods section is sufficiently detailed, but should be updated regarding the use of the spatial analyses (see details below). This will also require rerunning the analyses.
Another fundamental issue is that no global test of significance was performed, in the simulations, before performing a forward selection, while this is has been clearly shown to be crucial to avoid inflated type I error rates (Blanchet et al. 2008, Bauman et al. 2018a). The inclusion of this step will be necessary in the study. Indeed the forward selection with double stopping criterion (Blanchet et al. 2008) begins with a global test of significance (test of the response data against the whole set of predictors), and the selection is run if and only if this global test is significant.

Overall, the study has a good potential, but some substantial additional work is still needed, regarding the experimental design, for the authors to generate robust and reliable results related to up-to-date ways of performing eigenvector-based spatial analyses, and to draw their final recommendations.

Experimental design

Line 56: Add a coma after “dispersion”.

Lines 64-67: I see several issues in this paragraph.
The PCNM have been replaced by the Moran’s eigenvector maps (MEM) in 2006 (Dray et al. 2006), more powerful, more flexible, and straightforward with respect to the sign of the spatial autocorrelation (see Dray et al. 2006 for details).
Although PCNM have still been used since then, they should have not been (see previously mentioned reasons) and should have been replaced by MEM, as it was the case in most studies published after 2006. This has recently been shown and discussed in two recent papers (Bauman et al. 2018a, b, Ecography, Ecology).
Regarding the selection of a subset of spatial eigenvectors, Blanchet et al. (2008, Ecology) showed that the forward selection presented a correct type I error rate as long as it was preceded by a global test (i.e., test of the response variable or matrix against the whole set of explanatory variables). Moreover, the correct type I error rate of this forward selection usually applied to spatial eigenvectors has recently been confirmed for different sampling designs and types of response variables (Bauman et al. 2018a, Ecography). This last study, as well as the one by Bauman et al. (2018b) both showed that not only does the forward selection of Blanchet et al. (2008) have a correct type I error rate, but it also yields accurate (and non-overestimated) estimations of the R². Bauman et al. (2018c, Oikos) recently discussed the results obtained by Gilbert and Bennett (2010) and cited by the authors.
This background introduction part therefore needs serious updates from recent advances to the field (considering MEM instead of PCNM, and optimization of the selection of a spatial weighting matrix as well as a subset of spatial eigenvectors, see references above).

Line 97: The correct term would be “extent”, instead of “scale” here (see Legendre and Legendre 2012, or Scheiner 2011, Frontiers of Biogeography).

Line 98: Again, to me, there is a fundamental problem in trying to compare the performance of two methods coupled with an outdated technique (PCNM). The authors should redo the analyses using MEM instead of PCNM, and ideally relating the selection of spatial eigenvectors to Bauman et al.’s review and study on this subject (both in the introduction and in the analyses), as these studies already solved some of the issues presented in the manuscript. If the authors decide to do otherwise, then they should justify their choice.

Lines 108-118: It would be useful to display a map of the three datasets, to visualize their sampling designs for example. This could go to the supplementary material.

Line 120-121: See above. Choosing PCNM instead of the correct up-to-date practice (since 2006) which is to use the more flexible and powerful MEM, is a fundamental problem for a study aiming at recommending good practices in terms of spatial analyses when dealing with count data. If the article aims at giving recommendations for future studies, it needs to be based on the up-to-date and optimized way of performing the concerned spatial analyses.

Line 121-123: This claim is correct, but again, it is out of date, as MEM are much more flexible and powerful (e.g. they can be built from graph-based spatial weighting matrices that yield more powerful and accurate results for the very common irregular sampling designs, see Bauman et al. 2018b).

Lines 125-127: If PCNM were computed following Borcard and Legendre (2002, Ecological Modelling), as stated here, then the positive eigenvalues do not correspond to positively autocorrelated spatial structures only. This straightforward link stands only when using MEM (this is in fact one of the plus of the MEM with respect to the PCNM, see Dray et al. 2006 for details).

Lines 136-137: This could easily be done using MEM. Then, the authors will be allowed to consider all positive spatial eigenvectors as related to positively autocorrelated patterns (see above). I would suggest to generate MEM variables (i.e. spatial eigenvectors) from a suited spatial weighting matrix (depending on the sampling design, see Bauman et al. 2018b), and to make the selection of spatial predictors within the positively autocorrelated eigenvectors, as already done with the PCNM in the current version of the manuscript.

Line 203: The MEM variables can easily be generated using the package adespatial. The function listw.candidates() allows easily generating one or several spatial weighting matrices. The function listw.explore() can be useful to visually define apriori the spatial weighting matrices that could be suitable choices. The function scores.listw() can then be used to generate all the spatial eigenvectors associated to positive autocorrelated structures, for example. The function mem.select() allows testing the significance of the spatial weighting matrix, and if and only if the latter is significant, then it can use different ways of selecting a subset of spatial predictors (see Bauman et al. 2018b for details).

Lines 214-215: The forward selection of Blanchet et al. (2008) consists in (i) performing a global test of significance, and (ii) to proceed with the forward selection with double stopping criterion if and only if the global test is significant. This is half of the message of their study, as this is the step which avoids a hugely inflated type I error rate. And this, since then, has been the correct way of performing the forward selection, following their study.
From what I read here, it seems that the authors only used the double stopping criterion for the forward selection (which is good and important to avoid overestimation of the R-squared), but did not first check the significance of the global model (i.e. including all explanatory variables) to decide whether to perform the variable selection, or not. If this is so, then their comparison of this practice to the other is expected to be strongly biased, at least in terms of type I error rate. This is an important point to correct.
I recommend correcting this biased way of using Blanchet et al.’s forward selection. The authors can, additionally, keep their current analysis on the forward selection without a global test, to highlight the need of the global test beforehand. However, since Blanchet et al. (2008) have already extensively showed in their simulation that this practice yields biased results, I feel this would only make the article more complicated without any actual added value.
If a global test was already performed as a condition to perform the forward selection, then this should be clearly stated here. If it was not, then see above, and the analyses should then be rerun.

Line 220: This function is now part of the package adespatial (Dray et al. 2018).

Lines 223-224: Just an idea here, but the genetic algorithm of Calcagno et al. (2010, Journal of Statistical Software; package glmulti) might be a nice solution to this limitation.

Validity of the findings

This section will depend on the new results, once the methodological improvements will have been implemented (see previous sections).

Additional comments

Lines 305-308: The present study is about selecting a subset of spatial eigenvectors when working on binary presence/absence data. In their recent review on a very related topic, Bauman et al.’s (2018a) compared different commonly-used methods of spatial eigenvector selection in terms of type I and II error rates, for count data notably. They highlighted that, when correctly used (see comment above), the forward selection (Blanchet et al. 2008) following a global test of significance yielded the highest accuracy and statistical power, even though it often selected a few extra explanatory variables. Their study, in addition, showed that the eigenvector-selection method should ideally be guided by the objective for which the spatial predictors will be used (that is, controlling for spatial autocorrelation, or capturing and describing the spatial structures as accurately as possible).
The present study would need, ideally, to discuss their results in the light of this recent study, or should at least relate to it.

Moreover, I wonder whether the overall accuracy is not better grasped by using an overall adjusted R-squared (as in Bauman et al. 2018a, b), instead of only accounting for a proportion of well or mis-selected explanatory variables, as a wrongly-selected explanatory spatial variable could have a coefficient so small that it would not actually influence the overall detected pattern and its R-squared, for example. I am not saying that one approach is fundamentally better, but that these seem to be complementary components of the accuracy, and that it would be an informative additional statistics to provide more robust results regarding the accuracy comparison, and hence better support the conclusions of the study.

Distance-based spatial weighting matrices (which includes PCNM) have been shown to lack statistical power and to yield inaccurate estimates of the R-squared when working on irregular sampling designs (Bauman et al. 2018b). Since the sampling design is irregular here (at least for some of the datasets of the study), not only would it be more adapted to use a graph-based spatial weighting matrix to retrieve the spatial patterns of the simulated data, but it mostly raises the question of the selection of a spatial weighting matrix. The latter topic is a key step before selecting a subset of spatial eigenvectors (e.g. Dray et al. 2006, Stakhovych et al. 2008, Bauman et al. 2018b). This subject should at least be discussed somewhere, as it will relate directly to the final subset of selected spatial eigenvectors.

Line 312 (and general): The authors talk of “type I error rate” for the selection of each given spatial variable, which, in my opinion is somewhat misleading, since “type I error rate” generally refers to the detection of an effect where there is none. So, the type I error rate of the global test would make sense, as it would indicate a proportion of cases in which a spatial effect was detected in non-spatially structured data. However, what can we learn from “type I error rates” for each spatial variables, when there is an overall spatial effect, as the coefficient of a wrongly-selected spatial variable could be very small or very large.

Lines 353-358: This relates directly to the minimization of the spatial autocorrelation in the model residuals, using a minimum number of spatial predictors, as method of spatial eigenvector selection (Griffith and Peres-Neto 2006, see “MIR approach” in Bauman et al. 2018a). The authors could include this into the discussion to better relate to recent studies on the subject.

Line 361 (and related to the overall spatial methodology): One of the reasons why distance-based spatial weighting matrices (and therefore PCNM) are not as suitable for irregular sampling designs is that a distance threshold is used to define the connectivity matrix (matrix B, see Dray et al. 2006). This distance threshold defines which sites are connected and which ones are not, and is most often (as in the PCNM) the largest edge of a minimum spanning tree (i.e. the smallest link allowing all sites to remain connected). As a consequence, even only one site more isolated from the other sites is enough to increase this threshold distance, and, as a consequence, causes some distant sites to be connected (see details in Bauman et al. 2018b). This decreases the ability to model fine-scale spatial patterns, so that I would suggest using a graph-based spatial weighting matrix to build the MEM variables of the simulation study, in order to be able to design actual fine-scale spatial patterns, regardless of any limitation caused by a distance-based connectivity threshold.

Lines 371-376: The conclusion may need to be adapted, once the new analyses will have been performed.

References

Bauman et al. 2018a. Disentangling good from bad practices in the selection of spatial or phylogenetic eigenvectors. Ecography 41(10): 1638-1649.
Bauman et al. 2018b. Optimizing the choice of a spatial weighting matrix in eigenvector-based methods. Ecology 99(10): 2159-2166.
Bauman et al. 2018c. Testing and interpreting the shared space-environment fraction in variation partitioning analyses of ecological data. Oikos: doi: 10.1111/oik.05496.
Blanchet et al. 2008. Forward selection of explanatory variables. Ecology 89(9): 2623-2632.
Borcard and Legendre 2002. All-scale spatial analysis of ecological data by means of principal coordinates of neighbour matrices. Ecological Modelling 153: 51-68.
Calcagno et al. 2010. glmulti: An R package for easy automated model selection with (Generalized) linear models. Journal of Statistical Software 34(12): 1-29.
Dray et al. 2006. Spatial modelling: a comprehensive framework for principal coordinate analysis of neighbour matrices (PCNM). Ecological Modelling 196(3): 483-493.
Dray et al. 2018. adespatial: Multivariate multiscale spatial analysis. R package version 0.3-2. [https://cran.r-project.org/web/packages/adespatial/index.html]
Griffith and Peres-Neto 2006. Spatial modeling in Ecology: the flexibility of eigenfunction spatial analyses. Ecology 87(10): 2603-2613.
Legendre and Legendre 2012. Numerical Ecology, Elsevier, Amsterdam.
Scheiner 2011. Musings on the Acropolis: Terminology for biogeography. Frontiers in Biogeography 3: 62-70.
Stakhovych et al. 2008. Specification of spatial models: A simulation study on weights matrices. Papers in Regional Science 88(2): 389-408.

---

## Round 0.2 · Major Revisions

Thank you for your major effort to overhaul this manuscript. As you will see, Reviewer 3 has again reviewed and provided excellent feedback to improve the manuscript. I would very much like to accept this manuscript in a revised format addressing Reviewer 3. Many of the comments are straight forward to address and I see the effort invested in the feedback as proof positive this paper is worth reading. Please work through these comments and resubmit. A very nice first set of major improvements and I am pleased you were able to resolve the code error.

Reviewer 3 ·

Basic reporting

* The introduction and discussion of the manuscript have substantially been improved on the basis of some of the previous comments and suggestions by me and the other reviewers.
The authors changed their use of the outdated principal coordinates of neighbourhood matrices (PCNM) for Moran’s eigenvector maps (MEM), that any study aiming to study beta-diversity with eigenvectors should use. This has significantly improved the approach used in the manuscript.
The authors also explicitly expressed their reasoning for their choices of spatial weighting matrices (SWM) for their three datasets, following recent recommendations maximising statistical power and ensuring an unbiased approach of the spatial component of the data.
The study now also includes a global test of significance before carrying on a method of variable selection. This was a central issue in the previous version and was addressed in the resubmitted version.
The authors also added maps of their datasets in supplementary, which eases the understanding and visualisation of their case studies.

* However, there remain some major unresolved issues in the manuscript (most of them already raised in the previous review).

Major comments:
* * *
* An important issue in the message of the article regards the way the authors chose to address the performance of the two approaches they compare (in spite of the fact that this was already raised by me and other reviewers in the previous review). The performance is only assessed using a proportion of correctly selected spatial eigenvectors, while these are only tools used highlight spatial structures through their linear combination, and are not used separately.
A related issue is the misleading and erroneous use of the term “type I error rate”, with respect to what is actually tested.
The type I error rate, in the frame of this work, would be how frequently a spatial signal is detected when there is none. The authors confound this aspect – which is not the focus of the paper nor is tested here – with the identity of the spatial predictors selected by the selection methods that they test, and their number. This confusion appears from the abstract (line 45: “retaining too many explanatory variables leading to high Type I error rates”) to the manuscript at different places.
In addition, what matters when studying the determinants of community beta-diversity and using spatial eigenvectors is (1) that a spatial pattern is not wrongly detected when there are no spatial pattern in the process generating the binary data (real type I error), and (2) when there is a spatial pattern in the response data, that the influence of the spatial pattern on the response is correctly assessed both quantitatively (adjusted R-squared for instance) and qualitatively (visual look of the pattern).
Regarding the point (1), I don’t think that the type I error rate of MEM variables has been tested on presence-absence data in combination with a forward selection to select a subset of spatial predictors, but given the global test of significance and the double stopping criterion of the method, no type I error rate inflation is to be expected (this would therefore not add much to the manuscript). Regarding the point (2), the forward selection with double stopping criterion used on spatial eigenvectors (MEM variables) has already been shown to sometimes select too many explanatory variables in some instances (Blanchet et al. 2008, Bauman et al. 2018 – Ecography), but does not inflate the actual contribution of the spatial predictors (i.e. adjusted R-squared; Blanchet et al. 2008, Bauman et al. 2018 – Ecography, Ecology). Moreover, the spatial predictors selected by a selection method are not important themselves, as they only are the mathematical tools that, when combined properly through a linear combination, will reveal the spatial patterns present in the response data. Therefore, as long as the patterns obtained through the model (linear combination) are correct, the exact identity of the tools used to generate the final pattern (the identity of the MEM variables) do not matter so much. Some consecutive MEM variables, for example, can display very similar patterns, so that visually assessing the actual spatial pattern resulting from the linear combination of the spatial predictors, could as well show that selecting a few MEM variables too many but with small coefficients does not significantly modify the overall patterns in the community.
However, the authors do not assess the spatial patterns resulting from the combination of the selected spatial eigenvectors – be them those used to simulate the data or not. This is a central issue of the manuscript in its current state. I would therefore suggest correcting the misleading use of “type I error rate”.
A few visual assessments (maps) comparing the true spatial pattern (the one the model aims at recovering) with the spatial pattern obtained through RDA/FW and GLM/AIC would be important too, in order to assess the issue raised above. It would additionally help assessing whether the excess of MEM variables selected have real consequences or not. I expect this to (1) give the manuscript enough robustness and ground to achieve a sufficient confidence for the recommendations it aims to provide, and (2) refine the discussion by weighting correctly what aspect of the analysis is affected. This is also a more thorough and robust way of comparing the potential lack of power/bias related to RDA/FW for binary data, but could as well show that there are little differences between the two approaches. The paper aims to provide very general and important recommendations for many fields of ecology. Assessing a proportion of spatial predictors wrongly selected as sole way of comparing two approaches may be insufficient for that, without assessing the consequences in terms of spatial pattern modelled by the combination of the selected variables with their respective coefficients.
Without this, a strong motivation to recommend the use of GLM/AIC instead of RDA/FW is lacking, as the fact that the FW selects more predictors than the AIC has virtually no consequences on the quantitative effect of the spatial predictors, and possibly no real effect on the precision of the patterns themselves.

* I would recommend English proofreading by a native speaker.

Minor comments and corrections:
* * *
Minor comments and corrections:

* Figure 1: Problem with spaces missing or too many both in the figure and in the legend.
Matrix X doesn’t appear clearly in the figure. I would add a step with a matrix of selected spatial variables to make this clearer and more explicit. After all, the step of MEM variable selection is the centre of what is assessed in the study.

* Table 1 and its legend need some improvement to be made clearer, more intuitive and appealing. Better labels for the rows and columns would greatly help, and the legend could be improved too.

* Line 29: “estimating” is a little ambiguous. Change for describing or quantifying?

* Line 33: Rephrase to be clearer. What part of the datasets was used? The spatial coordinates? The original presence-absence data? The “with their respective sets of eigenvector…” gives the impression that the MEM variables are part of the datasets while they were generated by the authors. Rephrase to make clear what comes from the dataset and what was generated. “eigenvector-based spatial descriptors” is a little strange, and could be changed for “spatial eigenvectors”, “spatial eigenvectors used as spatial predictors”, etc (the spatial descriptors are not eigenvector-based: they are eigenvectors).

* Line 39: Add “The” before “Performance”. Correct this lack of article through the manuscript.

* Line 45: “We found that THE spatial scale OF THE PATTERN” (add the upper case words).

* Line 61 and 64: Coherence in the citation style.

* Line 65: Remove “a model of”. The eigenvectors are not generated from a model, but from a matrix of (weighted) connectivity among the sites.

* Line 80: “between THE response VECTOR OR MATRIX” (add the upper case words). “and all THE” (add the article).

* Line 81: Remove “possible relevant”. This sentence aims to explain the global test performed before performing the selection. Choosing relevant explanatory variables comes before a global test, as only variables making biological or ecological sense should be entered in the global model anyway. I would suggest to not mix this different aspect with the focus of the sentence to keep it as clear as possible. If the authors want to keep this aspect, I would suggest a separate sentence (but this is not necessary in the frame of this work, to me).

* Line 84: This is a detail, but the significance level should come first, as it is the first stopping criterion (both historically and in the method).

* Line 89-91: This aspect is at the centre of the motivation and expectations of the paper and could therefore be further explained.

* Line 93: Remove the “not” after “always”.

* Line 97: “in a natural way” is strange. In a statistical way using the probability distribution of maximum entropy? See McElreth 2015 (Statistical Rethinking) for more details or ideas.

* Line 98: “has long been established” instead of “has been long-established”

* Line 99: “analyses” instead of “analysis”. And change “there are now” by “and”.

* Line 101-102: What the paper compares is not the RDA vs GLM, but the variable selection method used with both methods.

* Line 105: The authors could be more specific here. “the performance” is vague, mostly considering that the article only focuses on the proportion of variables correctly selected (and performance could refer to the true type I error rate, statistical power, quantitative accuracy like an adjusted R2, etc).

* Line 145: “generates THE SPATIAL WEIGHTING MATRIX (matrix W)” (add the upper case words and the parenthesis).

* Line 151-153: Unnecessary details for the manuscript. I would suggest removing this sentence. Moreover, the second function would be listw.select(), not scores.listw(), and listw.candidates() would be missing to talk about “from a range of B and A combinations”.

* Line 173: adespatial: The year of this version of the package is 2019, not 2018. Also, Bauman D is missing from the authors.

* Line 179: Again, the “with their respective original sets of positive MEMs” should be written differently to make clear that the MEM variables are not part of the datasets (or past papers were they were presented). And specify what part of the datasets was used to avoid confusion.

* Line 179: Remove the “then”, and begin the sentence with “To do so,”, as this does not come after what was done in the previous sentence, if I understand correctly, but provides details about the sentence.

* Lines 181-184: This part is difficult to read and understand, and so are the corresponding legends of the associated figure and table (as well as the table itself). Please improve the structure of the paragraph and provide clearer explanations of the simulation process.

* Line 186: Matrix B is already used to describe the connectivity matrix for the construction of the spatial weighting matrix. Choose another symbol and adapt in the figure too.

* Line 193: Lack of an asterisk after “B”.

* Line 194: Replace “were” by “are”.

* Line 196: Lack of spaces at several places.

* Line 200: Replace “in which” by “to which”.

* Line 206: “structureS” (add the “s”).

* Line 228: “sampling-RELATED absences” instead of sampling absences?

* Line 240: The authors do not compare GLM and RDA variable selection, but FW and AIC used with RDA and GLM, respectively.

* Line 242: Terminology issue: Be sure to be consistent through the whole manuscript using “broad” as opposed to “fine” spatial scales, but avoid using “small” and “large” scales, because of their counter-intuitive use in geography (see Legendre and Legendre 2012, Numerical Ecology, for details). A small scale in geography would correspond to a broad scale in ecology, and a large scale in geography corresponds to a fine scale in ecology. The terms broad and fine scale should therefore preferably be used in ecology to avoid confusion.

* Line 243: Suggestion, for clarity: “number of MEM variables creating the spatial structure in the data (i.e. having non-zero coefficients)”.

* Line 245: not “up to three”, but “three”, simply.

* Line 245: “spatial scaling patterns” sounds strange to me. I would simply say “spatial scale of the pattern” or would at least make explicit what is meant by “spatial scaling patterns”.

* Line 246: Change “large” in “broad”.

* Line 247: “B*” is sometimes bold and sometimes not.

* Line 257: Remove “orthogonal”: unnecessary as the orthogonality is an intrinsic property of MEM variables.

* Line 272: “THE forward selection WITH DOUBLE STOPPING CRITERION” (add the upper case words).

* Line 273: Different things are mixed in the sentence. I suggest: “We used the forward selection with double stopping criterion following Blanchet et al. (2008), beginning with a global test of significance (model with all spatial predictors), and carrying on with the variable selection if the global model was significant. The forward selection itself consists of a stepwise procedure including in the model the variable contributing the most to the R-squared. The procedure stops either when the next variable with the highest contribution is not significant (first stopping criterion) or causes the model R-square to be bigger than the R-square of the global model (i.e. containing all variables; second criterion).”
This would also replace the wrong explanation provided lines 277-278 (“the smallest p-value among the remaining excluded variables exceeds an alpha threshold”).

* Line 280: Remove “also” and “null”, and add “with intercept only” after “model”.

* Line 284: Add “The” before “Performance”.

* Line 286: Remove “Type I errors” (see previous comments).
* Line 286: Remove “Type II errors” (see previous comments).

* Line 296-297: The adjusted R2 is also called coefficient of determination.

* Line 320: Problem in the sentence. Suggestion: “suggests both frameworks are insensitive to how presence/absence data WERE modelled (either as a proxy for real community composition of as a sampling artefact)”.

* Line 323-324: Rephrase to avoid the terms type I and II error.

* Line 382: “corresponding” is ambiguous. Please be more specific.

* In the legend of Figure 1, change “Hellinger” in “Ochiai”. And correct the typos.

Experimental design

Major comments:
* * *
* A way making more sense to compare the two approaches for the objective of the manuscript would be to have actual environmental variables and MEM to control for spatial autocorrelation in the residuals. Indeed, ecologists are interested in environmental variables. Retrieving the correct coefficients for the environmental variables after controlling for spatial autocorrelation in the residuals (through a selection of spatial predictors; see MIR approach for example in Bauman et al. 2018 – Ecology) through RDA and GLM would be a more direct manner of defining the approach with the highest accuracy to retrieve the correct coefficients for the actual variables of interest.
If the manuscript focuses on recommendation for studies using environmental variables, this would be a better way of addressing the question. If it focuses on correctly detecting and describing the spatial patterns of a multivariate binary dataset, then the previous comment (previous paragraph) is even more important.

* The letter “B” is used in the manuscript for two very different things: the connectivity matrix used to build a spatial weighting matrix from which the eigenvectors are generated, and the matrix of coefficients for the spatial eigenvectors (line 186 etc). The latter should be changed both in the text, Fig. 1 and its legend into a different letter to avoid confusion.

* Line 41: Wrong use of the concepts of type I and type I error rates (see first comment).

* Line 44-45: Retaining too many variables is a different problem than type I error rate inflation (see Blanchet et al. 2008 for example). This confusion through the manuscript may give a misleading message where the reader may understand that spatial structure are wrongly detected when there are none in the data.

Line 291-293: From what I understand, this sentence does not make much sense, and is an incorrect parallel to what is stated by Fraleigh & Beauregard at the pages cited. I may misunderstand something, but went several times through these lines and thought about it, and still really do not think that this, nor what is stated from line 293 to 295, is correct. This is a central problem in the current state of the manuscript, as this is the only dimension used to assess the performance of the two approaches, which makes of this way of considering type I error a problem at the foundation of the results, discussion and recommendations of the paper. I detailed this point at the very beginning of the review (second paragraph).

The question and aim of the manuscript are interesting and potentially important if there is a strong difference between the two approaches. However, I feel that in its current state, the authors fail to correctly address the question in a robust and reliable manner. This is the central problem of the manuscript, in its current state.

In their answer to my first comment about the use of “type I error rate”, the authors answered the following: “This is similar to criticism from reviewer #1. Note that linear combinations will be unique for any given set of independent variables and therefore ca only be fully correct by the correct inclusion/exclusion of individual variables. Moreover, stepwise inclusion (or exclusion) of individual variables is the usual way these models are built, which makes it a good framework to evaluate them.”
However, different sets of MEM variables (close to one another and therefore with similar eigenvalues) can, depending on their coefficients, yield very similar spatial patterns, so that the way the authors assess performance is misleading and insufficient to draw robust and global recommendations. A second issue is that not only is this way of assessing performance insufficient, but it also makes little ecological sense, as the spatial eigenvectors are not used separately on real datasets, but through their linear combination to highlight complex patterns. Another use, as if not more common, is the use of the spatial predictors with environmental predictors, that the authors did not use in their comparison of two approaches that will be used with such environmental variables.

Validity of the findings

* All the data underlying the analyses as well as the R code have been provided.

Major comments:
* * *
* Line 369-376: The proportion of MEM variables correctly selected by definition is the product of the variable selection approach, not the RDA or the GLM. And even if it were the case, this is not was is tested or shown in the manuscript. This whole portion of text however focuses on discussing the RDA.

* Line 405: Type I error rate is a problem here again, as the authors compare what they call type I error (wrongly) to what is usually considered as type I error rate, in the other study.

* Line 421: Another example of the confusion arising from the wrong use of Type I error rate. The authors, here, use “type I error rate” (and “rate” is missing) in the correct sense of the term, while using it with a different (and unusual) definition throughout the whole manuscript.

* Lines 425-427: Regarding “it is also likely that…”: This is absolutely not sure, as it is very different to consider a high number of orthogonal variables covering the whole range of possible spatial patterns detectable with the sampling design, or a much smaller number of non-orthogonal environmental variables. This is precisely the reason why if the purpose of the manuscript is to provide recommendations for studies using environmental variables, then using both (e.g. non-orthogonal) environmental variables and MEM variables might be the way to go.

Additional comments

The authors put a lot of work to tackle some of the issues raised by the two other reviewers and me and this has improved the manuscript, mostly with respect to the initial errors in their code, the absence of a global test of significance before using the forward selection, and regarding the now up-to-date use of spatial eigenvectors.
The question and aim of the manuscript are interesting and potentially important if there is a strong difference between the two approaches. However, I feel that in its current state, the authors still fail to correctly address the question in a robust and reliable manner, in spite of the fact that this issue was already raised and emphasised by me and reviewer 1. This continues to be the central problem of the manuscript. I detail the different aspects of this issue in the review and made some (I hope, constructive) suggestions to solve this problem.
The article aims to provide very general recommendations regarding how key questions of community ecology should be addressed, and therefore needs to address this question thoroughly, which is not the case yet, according to me.

---

## Round 0.3 · accepted · Accept

Thank you for your serious efforts and good faith effort to work through the review process. As you will see from the response, the hard work has greatly improved the manuscript and the paper is ready for publication.

·

Basic reporting

The authors have made an impressive work and have clearly invested considerable efforts and energy to address all my previous comments. I read their responses to my concerns, suggestions and corrections, and appreciated that they took them seriously and responded thoroughly. I can see how much the manuscript has improved both in depth, precision in the terms used, and clarity in the objectives of the study, the methods, results (and figures) and discussion. I appreciate how much work and time this must have taken, but am happy it was done now that I read the new version of the manuscript and the track change version. I believe this will make the paper more accessible and understandable by a broader audience, hopefully reaching more of the targeted ecologists. Finally, I trust the results more after reading this version than before, and think the framing of the results and discussion is more correct and nuanced now, which I think improves the work.

Experimental design

-

Validity of the findings

-

Additional comments

-